# “I’ll Continue If I Have a Positive Mind”: Identifying the Ways in Which Depression and PTSD Impact PrEP Adherence Among PrEP-Experienced Pregnant and Postpartum Women in Cape Town, South Africa

**DOI:** 10.3390/ijerph22091350

**Published:** 2025-08-28

**Authors:** Amelia M. Stanton, Madison R. Fertig, Jennifer Nyawira Githaiga, Devisi A. Ashar, Linda Gwangqa, Melinda Onverwacht, Lucia Knight, Landon Myer, Jessica E. Haberer, John Joska, Conall O’Cleirigh, Christina Psaros

**Affiliations:** 1Department of Psychological and Brain Sciences, Boston University, Boston, MA 02215, USA; mrfertig@bu.edu (M.R.F.); daashar@bu.edu (D.A.A.); 2Division of Social and Behavioural Sciences, School of Public Health, University of Cape Town, Cape Town 7925, South Africa; jennifer.githaiga@uct.ac.za (J.N.G.); linda.gwangqa90@gmail.com (L.G.); monve1428@gmail.com (M.O.); lucia.knight@uct.ac.za (L.K.); 3School of Public Health, University of the Western Cape, Bellville 7535, South Africa; 4Division of Epidemiology and Biostatistics, University of Cape Town, Cape Town 7701, South Africa; landon.myer@uct.ac.za; 5Department of Internal Medicine, Massachusetts General Hospital, Boston, MA 02114, USA; jhaberer@mgb.org; 6Department of Medicine, Harvard Medical School, Boston, MA 02115, USA; 7Faculty of Health Sciences, University of Cape Town, Cape Town 7935, South Africa; john.joska@uct.ac.za; 8Department of Psychiatry, Massachusetts General Hospital, Boston, MA 02114, USA; cocleirigh@mgh.harvard.edu (C.O.); cpsaros@mgh.harvard.edu (C.P.); 9Department of Psychiatry, Harvard Medical School, Boston, MA 02115, USA

**Keywords:** pregnancy, pre-exposure prophylaxis, HIV, South Africa, depression, posttraumatic stress

## Abstract

Pregnant and postpartum people (PPPs) face heightened risk for HIV acquisition, yet depression and trauma-related symptoms can undermine adherence to pre-exposure prophylaxis (PrEP). To inform the development of a brief mental health-focused adherence intervention, we explored the impacts of depression and posttraumatic stress disorder (PTSD) symptoms on PrEP use among PPPs in Cape Town, South Africa. Twenty-eight PPPs with elevated symptoms of depression and/or PTSD and recent PrEP adherence challenges completed qualitative interviews. Six antenatal providers were also interviewed. Thematic analysis revealed three key findings with subthemes that deepen exploration of each theme: (1) depression and PTSD symptoms contributed to missed PrEP doses or late pickups by increasing doubt about PrEP efficacy, amplifying pill burden, intensifying avoidance and withdrawal (e.g., hypersomnia and disengagement from providers), and disrupting memory through rumination and emotional overload; (2) most PPPs preferred support from professional counselors, while a minority preferred informal support; and (3) intervention design considerations included aligning patient and provider goals, selecting between individual or group formats, and addressing integration barriers such as staffing and space constraints. Providers affirmed the need for embedded mental health support. Intervention strategies that increase PrEP knowledge and motivation while targeting emotional withdrawal, fatigue, and cognitive overload may improve adherence and reduce HIV risk in this population.

## 1. Introduction

HIV acquisition rates remain disproportionately high among pregnant and postpartum persons (PPPs) in southern Africa, where women and girls account for 62% of all new infections [1]. In South Africa (SA) specifically, 27.5% of peripartum persons are living with HIV [2]. PPPs have a higher HIV susceptibility than non-pregnant persons due to both biological and structural factors. Biologically, hormonal changes, alterations to the genital tract, and sexual behaviors of women and their partners contribute to heightened risk during pregnancy and postpartum [3]. In southern Africa, the risk of HIV acquisition per condomless sex act is 3–4 times higher during late pregnancy and postpartum [4]. Structurally, factors like resource limitations, gender inequity, lack of knowledge about partner’s HIV status, engaging in condomless sex, or having multiple sex partners exacerbate this vulnerability [5].

Consistent use of HIV prevention strategies, like pre-exposure prophylaxis (PrEP), can greatly reduce new HIV infections among PPPs in southern Africa. When PrEP is taken as prescribed, PrEP can reduce the risk of contracting HIV through sexual activity by up to 99% [6]. Projections suggest that if 80% of pregnant individuals in SA adhere to PrEP through pregnancy and breastfeeding, perinatal HIV transmission could be reduced by 41% by 2030 [7]. Despite PrEP’s proven effectiveness in the region, there are several challenges to PrEP adherence, such as misperceptions of HIV risk, HIV-related stigma, and lack of social and provider support [6]. In SA, 84% of women at their first antenatal visit are initiated on oral PrEP; however, a third of these pregnant women discontinue PrEP after a month, and overall adherence remains low during follow-up [8]. Postpartum people, in particular, are more likely to discontinue PrEP than pregnant people, underscoring the need for targeted strategies to support continued use through the first year post-delivery [8].

Mental health symptoms, including symptoms of posttraumatic stress and depression, also negatively impact PrEP adherence during the peripartum period [9]. In low- and middle-income countries (LMICs), one in four women reports depression during pregnancy and one in five post-delivery [10]. Depression can impair engagement in protective health behaviors, leading to reductions in self-efficacy, healthcare engagement, and overall self-care behaviors [11]. Depression can also lead to social withdrawal and isolation, significantly minimizing one’s social support—an essential factor in sustained PrEP use [12].

In SA, intimate partner violence (IPV) is a pervasive concern, with more than 20% of all women experiencing at least one act of physical, psychological, or sexual IPV during pregnancy, and about 25% during the first nine months postpartum, contributing to increased levels of posttraumatic stress disorder (PTSD) in this population [13]. These IPV experiences heighten vulnerability to PTSD symptoms, contributing to PrEP non-adherence [14]. Specifically, IPV and relationship inequality may create barriers to PrEP uptake and adherence at multiple levels. At the individual level, IPV and relationship inequality can restrict access to information, reduce self-efficacy, and instill fear of relationship loss; at the partner level, difficulties in disclosing PrEP use and limited decision-making power around reproductive health can impede adherence; and at the community level, lack of PrEP awareness, inequitable gender norms, and PrEP-related stigma further reinforce these barriers [15,16].

Symptoms of depression and PTSD, including withdrawal and avoidance, can significantly disrupt adherence to HIV prevention behaviors like taking PrEP consistently. Individuals with PTSD may avoid activities or spaces that trigger traumatic memories, while those with depression often disengage due to sadness, anhedonia, or apathy, impairing their ability to refill and take PrEP as prescribed [9]. These symptoms can also intensify other barriers to adherence, including low social support, internalized stigma, and reliance on avoidant rather than problem-focused coping [17,18]. Feelings of hopelessness and guilt may reduce help-seeking, while depressive symptoms, loneliness, and low perceived social support have been shown to mediate the link between stigma and poor adherence, underscoring how mental health affects both intrapersonal and interpersonal functioning [19]. In the postpartum period, the intersection of perinatal depression and structural challenges—such as limited transportation, minimal HIV prevention knowledge, and low resource availability—may further reduce healthcare engagement and discourage continuation of PrEP [20,21].

While connections have been drawn between the ways in which depression and IPV impact PrEP adherence among women in southern Africa, the specific ways in which diagnostic levels of depression and PTSD symptoms impact PrEP adherence during pregnancy remain unclear, as well as how to best address PrEP adherence challenges among PPPs with mental health symptoms in a low-resource setting. To address this gap, the purpose of this study is to qualitatively explore the mechanisms by which depression and posttraumatic stress impact PrEP adherence and persistence during pregnancy and assess intervention preferences to inform the development of an intervention to improve PrEP adherence and mental health symptoms during pregnancy.

## 2. Methods

### 2.1. Study Procedures

PPPs and antenatal care providers were recruited from three antenatal clinics outside of Cape Town to investigate the psychological barriers to PrEP adherence during pregnancy and breastfeeding. PPPs were approached by study research assistants while waiting in queue at the antenatal clinic. Research assistants briefly described this study to PPPs, and those who expressed interest in completing an interview were invited to speak to a research assistant in a private space at the clinic to be consented and screened for eligibility. Research assistants did not ask potential participants any personal questions while in queue to ensure confidentiality. Similarly, providers were approached outside of their working hours (i.e., lunch break and after clinic closed) by study research assistants to determine their interest in being interviewed. Those who were interested were brought to a private space to screen for eligibility and complete the consent process.

Eligibility criteria for PPPs were as follows: (1) aged 15 years or older; (2) pregnant (any gestational age) and presenting for antenatal care or postpartum (up to one year); (3) HIV-negative (per chart review); (4) recent PrEP initiation (<1 month ago) or PrEP adherence challenges, either documented (>2 weeks late to pick up PrEP refill) or self-reported (via three items assessing missed doses, adherence quality, and adherence frequency over the past 7 days); and (5) elevated symptoms of depression (defined as a score of ≥11 on the Edinburgh Postnatal Depression Scale (EPDS)) and/or PTSD (defined as a score of ≥31 with the PTSD Checklist for DSM-5 (PCL-5)) [22,23]. The EPDS is a 10-item measure originally developed to screen for depression during the postpartum period, but it has been shown to reliably measure depression in pregnant persons, as well as depression among PPPs in SA-based studies [24]. Scores on the EPDS range from 0–30, with higher scores indicating elevated symptom severity; scores greater than 11 have been associated with diagnostic levels of depression per clinician-administered structured diagnostic interview [25]. The PCL-5 is a 20-item measure developed to assess PTSD severity utilizing the DSM-5 PTSD diagnostic criteria. Scores range from 0–80, with higher scores reflecting elevated symptom severity; scores greater than 31 have been associated with diagnostic levels of PTSD per clinician-administered structured diagnostic interview [26].

Participants deemed ineligible were compensated for the time they spent with study staff during the screening process (50 ZAR), while those who were eligible and consented to participate continued to complete the socio-demographic questionnaire. Participants provided informed consent or assent. In SA, individuals aged 15 and older can legally make their own reproductive health decisions [27]. Participants aged 15–18 therefore provided assent rather than consent prior to their interviews. All interviews were conducted in private spaces in the clinics by trained research assistants, who spoke both English and isiXhosa. PPPs received 250 ZAR for their time and travel reimbursement.

Across the recruitment clinics, six providers expressed interest in serving as key informants for intervention development, all of whom completed the brief questionnaire and a semi-structured interview. Due to local regulations, financial compensation is prohibited, so providers received refreshments only.

Following the qualitative interview principles outlined by Huberman and Miles, [28] interviews with PPPs consisted of questions and probes to explore (1) the impact of mental health symptoms (i.e., depression and posttraumatic stress) on PrEP adherence during pregnancy and postpartum; (2) past, current, or anticipated barriers and facilitators to PrEP adherence during pregnancy and postpartum; (3) existing self-care skills and attitudes toward help-seeking for their mental health symptoms; (4) interest in participating in an intervention designed to improve PrEP adherence and mental health symptoms; and (5) intervention preferences (i.e., desired content, interventionalist preference, and length and timing of sessions). Questions and probes varied slightly depending on the participant’s pregnancy or postpartum status. Additionally, provider interviews included questions and probes to explore their perspectives on the ways in which mental health symptoms impact patients’ engagement in care, as well as anticipated barriers and facilitators to implementing a PrEP adherence intervention in the clinic. Both guides were co-created and jointly developed by our SA- and US-based teams.

### 2.2. Analysis

Participant interviews were audio-recorded, transcribed, and translated (when needed) from isiXhosa to English by SA-based research assistants (LG and MO). Qualitative analyses for these interviews followed the principles of thematic analysis [29]. The second author (MRF) read several transcripts and utilized inductive and deductive coding strategies to develop the codebook. Four US-based research assistants (MRF, KEK, JS, and DAA) then open-coded two transcripts and made revisions to ensure clarity and consistency throughout the codebook, which was then reviewed by the PI (AMS) and the SA-based project coordinator (JNG). Once the codebook was finalized, eight transcripts were double-coded (by MRF, KEK, JS, DAA, AB, DK, and RM) using Dedoose (version 9) software [30]. Team members met to resolve any discrepancies in coding conventions or code definitions during the double coding process. The remaining 20 transcripts were then independently coded by the US-based research assistants.

Provider interviews were analyzed using rapid qualitative analysis procedures, a method for quickly and efficiently analyzing qualitative data to inform practice or make key decisions [31]. Authors MRF and DAA reviewed two transcripts to create a summary sheet and matrix templates. The summary sheet was used to identify meaningful content across all domains queried in each interview, and the matrix template collated all of the content from the individual summaries so that commonalities were observed across the corpus. Authors MRF and DAA separately completed the summary sheet and matrix for two transcripts and then met to reach consensus; DAA then analyzed the remaining four transcripts. Cultural nuances in either participant or provider transcripts were discussed with the SA-based team to confirm accurate interpretation. Once all the transcripts were analyzed, AMS and MRF extracted relevant quotes and met to discuss prevalent themes and descriptive quotes.

We followed Tracy’s eight criteria for high-quality qualitative research to guide our analysis [32]. This study addresses a worthy topic given the elevated HIV risk among perinatal individuals in South Africa and the public health significance of PrEP adherence. We ensured rich rigor by co-developing interview guides with our South African partners to elicit meaningful data and applying structured thematic and rapid analytic approaches with a collaboratively created codebook and summary sheet. To promote sincerity, we held weekly meetings with the SA-based team to reflect on analytic decisions and address potential cultural blind spots of the U.S.-based team. Credibility was strengthened through team-based discussions of coding, constant comparison, and consensus-building to confirm that findings were grounded in participants’ narratives. We aimed for resonance by selecting compelling quotes that conveyed participants’ lived experiences and preferences for support. This study makes a significant contribution by identifying modifiable psychological barriers to PrEP adherence during the peripartum period. We upheld ethical standards through local ethics board approval (HREC REF 484/2022), privacy protections, and practices to minimize clinic disruption. Finally, we achieved meaningful coherence by aligning our research questions, methods, and interpretations with this study’s aims and existing literature.

### 2.3. Results

A total of 65 participants were screened for eligibility, and of those, 28 qualified for this study. Most common reasons for ineligibility included not meeting criteria for elevated depression or PTSD symptoms (95% of total ineligible, *n* = 35), not self-reporting adherence challenges (84% of total ineligible, *n* = 31), and no delayed PrEP pickup (78% of total ineligible, *n* = 29); these reasons were not mutually exclusive, with many individuals meeting multiple criteria for exclusion. The final sample consisted of 28 PPPs, all of whom identified as women and had an average age of 28.9 years (SD = 7.4). At the time of the interview, pregnant participants (*n* = 10) were, on average, 25.9 weeks pregnant (SD = 8.0), while postpartum participants (*n* = 18) were, on average, 9.4 weeks post-delivery (SD = 11.2). Per medical chart review, 16 participants had recently been initiated on PrEP (<1 month) and 8 had PrEP adherence challenges (>2 weeks late to pick up PrEP refill). Of the 28 participants, 23 participants had self-reported PrEP adherence challenges, 7 of whom were also late to pick up their PrEP refills. Of the 28 participants, 16 (57.1%) had elevated depression and PTSD symptoms, 11 (39.3%) had elevated depression symptoms alone, and 1 (3.6%) had elevated PTSD symptoms alone. See Table 1 for full demographic details. The six providers who completed an interview were all professional nurses, averaged 8.3 years (11.9) working in antenatal care, and all endorsed asking patients about their mental health during clinic visits. See Table 2 for providers’ full details.

Our qualitative analyses revealed several psychological pathways to compromised PrEP use. Theme 1 articulates the specific ways in which depression and PTSD symptoms (e.g., sadness, anhedonia, shame, anxiety or stress, difficulty concentrating, avoidance, and withdrawal) led to missed doses or delayed pickup of PrEP at the clinic. Symptoms of both disorders contributed to PrEP use challenges via (1) increased doubt of PrEP efficacy, stemming from PrEP misinformation and stigma; (2) a strong sense of pill burden and concerns about the energy and effort required to continue taking pills; (3) forgetfulness associated with “thinking too much”, (4) fear-related avoidance manifesting as hypersomnia; and (5) withdrawal from interactions with healthcare providers. Theme 2 narrows in on preferred forms of PrEP support, with subthemes indicating (1) greater interest in receiving professional, confidential mental health counseling to support PrEP adherence and improve emotional well-being; (2) with a minority of participants preferring to confide in a peer, close friends, or family instead of counseling professionals. Finally, Theme 3 explicates the factors that are critical to consider when designing a mental health intervention to support PrEP adherence during pregnancy and postpartum, including (1) shared recognition of mental health as a foundation for PrEP adherence; (2) selecting an appropriate session format (group based vs. individual); and (3) addressing the logistical challenges of clinical integration. Themes and subthemes are illustrated in Figure 1.

Depression and PTSD symptoms leading to PrEP non-adherence, subtheme 1: increased doubt due to PrEP misinformation and stigma. Among participants experiencing symptoms of depression and/or PTSD, a prominent mechanism through which these mental health challenges impacted PrEP adherence was increased doubt about PrEP efficacy, fueled by misinformation and HIV-related stigma. Sadness, hopelessness, and cognitive symptoms such as rumination and indecision often intensified uncertainty about whether PrEP would be effective in preventing HIV, particularly during pregnancy. One participant shared, *“When I was feeling sad, I would start doubting and questioning if PrEP would really prevent me and my baby from getting HIV. I would start thinking about all the rumors people spread about PrEP”* (postpartum, age 22). While this participant ultimately continued taking PrEP to protect her baby, others described how emotional distress and misinformation eroded their motivation.

Stigma also played a critical role. Several participants described feeling judged when swallowing pills in the presence of others due to the assumption that they were taking antiretroviral therapy for HIV treatment rather than PrEP for prevention. One participant explained, *“Some people assume that it’s ARVs and don’t know what the pills are for so to avoid drama, I don’t take it in front of people besides my partner”* (postpartum, age 23). A provider echoed this concern, noting that stigma and misconceptions could lead patients to hide their medication or skip doses to avoid conflict or mistrust in relationships: *“People assuming that the patient is hiding their HIV status by saying they’re taking PrEP instead of ARVs. So, they would hide their pills because of the stigma. Sometimes the partner would tell them to not take PrEP or else there’s no trust between them”* (provider, age 62). These accounts highlight how anticipated and experienced stigma, likely amplified by mental health symptoms, can undermine adherence.

Depression and PTSD symptoms leading to PrEP non-adherence, subtheme 2: strong sense of pill burden and concerns about the energy and effort required to continue taking additional pills. For many participants, the act of consistently taking PrEP felt physically and emotionally burdensome. A core theme across interviews was a sense of pill fatigue, that is, feeling overwhelmed by the effort required to maintain daily pill-taking amidst already diminished mental and emotional resources. For example, one participant explained, *“I will see once I have given birth but it’s not nice drinking pills every day. Especially if you’re someone that’s not used to taking pills regularly. So, I don’t want to lie and say I will continue because I don’t know”* (pregnant, age 21). This participant’s uncertainty reflects how daily pill-taking can feel especially daunting for those unaccustomed to chronic medication use, compounding existing emotional and physical fatigue. In addition, depression-related symptoms such as low mood, exhaustion, and a lack of motivation directly interfered with participants’ ability to adhere to their PrEP regimen. As one woman noted, *“Sometimes I would not take it [PrEP] because I was not in the mood for anything”* (pregnant, age 33), while another shared, *“I am very hurt, and that sadness caused me to even not taking my PrEP the last weeks of my pregnancy”* (postpartum, age 29). These accounts illustrate the ways in which mental health symptoms contribute to a heightened sense of pill burden that compromises PrEP adherence.

Participants also described a broader loss of motivation and a withdrawal from health-related behaviors during periods of emotional distress. For some, depressive symptoms appeared to disconnect them not only from their routine but from the very rationale for taking PrEP: *“Sometimes I would ask myself what’s the reason for taking these tablets when I don’t even want tablets… I would not see myself in the clinic facilities or taking tablets. Nah!”* (postpartum, age 27). This kind of cognitive–emotional disengagement reflects the interplay between hopelessness, anhedonia, and pill burden. Rather than seeing PrEP as a protective tool, some participants viewed it as one more demand in a context already defined by emotional and physical depletion. The following quoted text emphasizes the extreme contexts in which some participants find themselves: *“I am a victim of 33 rapes. So, I wasn’t only taking PrEP. I took a lot of medication…I feel nauseous and I don’t feel like taking my tablets…it doesn’t feel good”* (postpartum, age 27). This excerpt underscores how trauma, depression, and pill burden can interact to create a state of profound disengagement, in which self-protective behaviors like PrEP adherence become emotionally inaccessible or even aversive.

Providers offered similar observations, noting that depression and PTSD often interfere with basic self-care, suggesting that adherence to a preventive regimen would be even more challenging than maintaining activities of daily living. *“Those that have PTSD…normally do not even take care of themselves. So, it will not be doable or easy to take pills if you cannot take care of yourself”* (provider, age 35). Others highlighted the added burden of new motherhood and inadequate social support. One provider explained, *“Even though the partner is present, [he] works during the day, leaving her struggling taking care of the baby…She’s dealing with a newborn; she’s dealing with the problems she previously had before giving birth. So, those are some of the instances that cause them to fail taking PrEP”* (provider, age 28). From the provider perspective, this convergence of emotional strain, caregiving responsibilities, and lack of support underscores how PrEP adherence becomes deprioritized when women feel depleted or unsupported.

Depression and PTSD symptoms leading to PrEP non-adherence, subtheme 3: forgetfulness associated with “thinking too much”. A third pathway through which depression and PTSD symptoms disrupted PrEP adherence was through excessive rumination—described by participants as “thinking too much”—and the associated difficulty concentrating, which often led to forgetfulness. Mental preoccupation with other concerns, particularly interpersonal conflict, fear, and self-blame, displaced the mental and emotional bandwidth needed to maintain a consistent adherence routine. As one participant explained, *“It affected me a lot because I used to think a lot about other things rather than taking PrEP”* (postpartum, age 18). In many cases, participants understood the importance of PrEP but still found themselves distracted by the emotional toll of trauma, relationship stress, or the daily demands of motherhood.

These ruminative preoccupations made it easier to miss or forget doses, which in turn triggered guilt or fear, particularly regarding potential HIV transmission to the baby. One participant reflected, *“It [the baby being infected] will hurt me because obviously I will blame myself… although there would be that particular thing that has distracted me that time. That will cause me not to take my PrEP and at the same time that might worry me”* (pregnant, age 39). These ruminative cycles may deepen emotional distress and further compromise adherence. Similarly, forgetfulness not linked to “thinking too much”, such as failing to bring PrEP when staying at a partner’s home for the weekend, was frequently described in the context of broader emotional or cognitive disorganization: *“Sometimes I forget taking my pills with [me] when alright with m… which means I end up not taking my pills for the whole weekend”* (pregnant, age 21). These accounts highlight how both persistent overthinking and more general forgetfulness, often shaped by underlying mental health struggles, can interfere with consistent PrEP use during pregnancy.

Providers echoed these experiences, noting that mental health symptoms could impair attention and short-term memory, particularly among individuals managing the psychological aftermath of trauma. *“Because of their mental state, they might forget to take it. Once someone has depression [they are] unable to think straight”* (provider, age 61). Another provider highlighted the cognitive burden of living with intimate partner violence: *“They are overwhelmed with fear and cannot think properly…they forget for some days, or they do not see it as a need… the only help they want is concerning their abuse”* (provider, age 35). In this way, trauma and mental health symptoms divert attention away from preventive health behaviors, including PrEP, by monopolizing emotional energy and narrowing perceived priorities.

Depression and PTSD symptoms leading to PrEP non-adherence, subtheme 4: fear-related avoidance manifesting as hypersomnia. PrEP non-adherence was often facilitated by an avoidance-based coping strategy: withdrawing from daily life through excessive sleep. Sleep, described not just as rest but as a way to escape or numb distress, became a form of psychological protection from overwhelming emotions. However, this strategy also disrupted medication routines, leading to missed or delayed doses of PrEP. For many, hypersomnia appeared to serve a dual purpose, providing relief from negative mood states and also avoiding interpersonal conflict: *“Yes, I prefer to sleep so that I do not come into contact with other people…so that they cannot be affected by my bad mood or else bursting out on others”* (pregnant, age 28). In these instances, sleep was used both to cope with inner turmoil and to avoid harming others, underscoring the self-protective and socially protective function of withdrawal. However, this withdrawal also extended to health-promoting behaviors such as taking PrEP. The same participant explained, *“I just feel like sleeping and do nothing at all. Yes, I sleep, and most of the things that I normally neglect even myself”*.

Disrupted sleep routines (i.e., too much sleep and, in some cases, disturbed sleep) also contributed to poor medication adherence due to timing challenges or a general sense of inertia. For example, one participant articulated, *“Sometimes when I have stress… I will say no, today I am tired I can’t do this. Because you know you didn’t take the pills. I will just avoid like that… let us do it tomorrow”* (pregnant, age 16). In this way, both depression-related lethargy and stress-induced procrastination interrupted daily adherence. Similarly, for some, emotional exhaustion evolved into full withdrawal from daily functioning: *“I become powerless, I lose interest in everything… I end up falling asleep. When I get out of that long sleep, I feel alright”* (postpartum, age 32). Here, sleep was described almost as a reset button, a way to recover from the weight of psychological pain, but it came at the cost of adherence to behaviors that require daily consistency.

Depression and PTSD symptoms leading to PrEP non-adherence, subtheme 5: withdrawal from interactions with healthcare providers. Depression and trauma-related symptoms also led to withdrawal from providers, which in turn disrupted participants’ engagement with care and contributed to missed PrEP pickups. This withdrawal occurs in the larger context of social withdrawal that is associated with depression and PTSD. Several participants described an overwhelming reluctance to speak with clinic staff, driven by emotional fatigue, irritability, or fear of being misunderstood. One participant shared, *“The other thing is that when I went to fetch my PrEP I was forced to speak because there were questions that I needed to answer. Then I was bound to answer those questions. So, I shouldn’t go”* (pregnant, age 28). In this case, even routine clinical interactions felt too demanding, and avoiding the clinic altogether was seen as a way to preserve emotional energy and avoid distress.

Negative encounters with providers, whether real or anticipated, further exacerbated participants’ feelings of sadness and withdrawal. One participant explained, *“I used to be alright. Not unless my sadness was caused by the nurse maybe here at the clinic”* and went on to clarify, *“I become disturbed and sad…I just become quiet and withdrawn. I feel sad”* (postpartum, age 29). These encounters contributed to a cycle in which interpersonal stress worsened emotional symptoms, which in turn reduced motivation to attend clinic visits or engage meaningfully with staff. Another participant reflected on how this cycle played out in her interactions with providers: *“So, when I am stressed, I do not feel like talking…So, I prefer to keep quiet, and they got more irritated…we would not understand each other, and I would have ended up not being helped”* (postpartum, age 20). Here, the participant’s attempt to avoid conflict through silence led to further misunderstandings and frustration on both sides, highlighting how trauma symptoms can interfere with even basic healthcare communication.

While most participants described withdrawal as a barrier to care, a few noted that feeling safe and emotionally supported in the clinic could shift this pattern. One participant explained, *“When I’m around people I tend to loosen up. Yes, I do talk about my feelings, and I do interact with [nurses] in a good way. I don’t lose myself”* (postpartum, age 27). This suggests that, for some PPPs with depression or PTSD symptoms, positive provider interactions may temporarily alleviate withdrawal and foster engagement in care. However, such interactions appeared to be the exception rather than the rule, suggesting the need for trauma-informed, empathetic clinical environments that proactively account for and accommodate the withdrawal tendencies common among people experiencing depression and/or PTSD.

Preferred forms of PrEP support, subtheme 1: Most participants were interested in receiving professional, confidential mental health counseling to support PrEP adherence and improve emotional well-being. Given the significant mental health challenges described, most participants expressed a strong interest in receiving professional psychological support. For many, counseling was seen as a potentially valuable avenue to better cope with the emotional toll of depression, trauma, and the daily stressors associated with pregnancy, postpartum recovery, and HIV prevention. One participant shared, *“I think it will be helpful because right now I don’t know how to cope on my own”* (postpartum, age 22), highlighting how overwhelming emotional distress had become and the need for structured, external support. Participants emphasized that the appeal of counseling was rooted not only in a desire to feel better emotionally but also in the opportunity to speak openly without fear of judgment. One woman explained, *“I will be free and will be easy for me to speak without feeling being judged so that the person can be able to advise me. How to go about in life so that my life can go alright”* (pregnant, age 33). This comment illustrates the perceived potential of counseling to offer both emotional relief and practical guidance, especially for those who may feel isolated or unsure about how to move forward in their lives. Crucially, some participants made a direct link between mental health support and improved PrEP adherence. One participant articulated, *“It would be a safe option because at the end of the day PrEP is something I would want to take. Having someone to talk to and help you take PrEP would be a good thing”* (postpartum, age 27). This suggests that mental health services could not only alleviate emotional symptoms but also enhance adherence to HIV prevention strategies by fostering motivation, accountability, and emotional resilience.

While many participants reported turning to friends or family members during times of distress, some emphasized the unique benefits of formal counseling. Trust and confidentiality emerged as key aspects of professional counseling relationships. For example, one participant stated, *“I for one would prefer to speak about it to someone… That would be a person that I have the confidence to trust… Maybe she can provide a manner where I can be able to forget about this thing so that my stress level can be minimized… I would like to go for counseling”* (pregnant, age 19). Therefore, the therapeutic relationship itself was seen as beneficial because of its neutrality and emotional safety. Similarly, another participant noted, *“What I like about counselling is that the person does not know you and your background. So, she will listen to me as a neutral person. I will feel free to speak freely and pour out my heart”* (postpartum, age 29). For individuals facing stigma, trauma, or interpersonal difficulties, this sense of emotional distance may be perceived as protective and enabling of deeper disclosure. Participants acknowledged that informal support networks could sometimes fall short; friends might be unavailable, or their guidance might not be adequate in moments of crisis. As one participant explained, *“I think it would be helpful because sometimes you go to a friend to talk and they’re not available… Also, counsellors are experienced. You might feel like giving up, but they are there to help you get through that, they know what to do in such situations”* (postpartum, age 22). This underscores the perceived reliability and expertise of counselors, especially in moments when mental health challenges intensify and when clear, constructive support is needed.

Even those who had never previously accessed counseling expressed curiosity and openness. For example, one participant explained, *“I have never attended counselling sessions so I don’t know what they usually look like, but I would like to get help on how to stress less and not think too much or how to stop doubting myself”* (pregnant, age 21). This sentiment reflects both a gap in mental health service utilization and an unmet need—participants recognized the impact of ruminative thoughts and low self-esteem on their well-being and were eager for tools to manage these challenges.

Providers also recognized the value of mental health support in improving both psychological outcomes and HIV prevention behaviors. One provider noted, *“They also get motivated after counselling when you make them realize that they don’t know their partner’s status or what they might be up to. It brings them fear of contracting HIV and reason to prevent”* (provider, age 62), indicating that counseling can be a catalyst for insight and behavior change. Another stated plainly, *“If the patient must see a social worker or a psychologist or even the counsellor for some reasons. That is the only support. I think that is the only support that we have for them”* (provider, age 35), emphasizing the critical need for integrating these services within existing systems of care.

Preferred forms of PrEP support, subtheme 2: A minority of participants would rather confide in a peer, close friends, or family instead of counseling professionals for mental health support. While most participants expressed a strong interest in professional counseling, a minority of participants preferred to confide in close friends or family members for mental health support. These participants emphasized the emotional safety, trust, and shared lived experience that such relationships could provide, benefits they felt might be harder to replicate in clinical settings. The value of peer connection and mutual understanding was emphasized by this participant: *“No, preferably a peer because there can be a mutual connection you see. Because we are sharing the same sentiment and understand what one is going through”* (postpartum, age 31). This perspective suggests that empathy and shared experience may be as important as formal training when it comes to choosing a source of mental health support. Peers who have navigated similar challenges were seen as particularly well positioned to validate experiences and offer practical, relatable advice.

For some, the preference to forgo professional counseling was rooted in logistical barriers or a lack of awareness about available mental health services. One participant shared, *“No, because I do not even know where to go for it and how accessible is it (counselling)… I think I am alright with my sister… It is because I trust her, and we grew up together”* (pregnant, age 23). This comment reflects how accessibility concerns intersect with the desire for familiar, emotionally secure relationships. For participants like this, a trusted family member offers a ready and reliable form of emotional care in the absence of clearly accessible formal services.

Intervention considerations, subtheme 1: shared recognition of mental health as a foundation for PrEP adherence. Participants and providers alike highlighted that any mental health intervention designed to enhance PrEP adherence must explicitly aim to address the underlying psychological symptoms—such as stress, anxiety, and trauma—that interfere with consistent pill-taking. The emotional distress many pregnant and postpartum women experience was seen as not only harmful on its own but also as a key barrier to engaging in health-promoting behaviors, including taking PrEP consistently. Participants emphasized the need for a nonjudgmental, supportive space in which to process their mental health challenges. One participant noted, *“I will be free and will be easy for me to speak without feeling being judged so that the person can be able to advise me. How to go about in life so that my life can go alright”* (pregnant, age 33), highlighting the importance of emotional safety as a precondition for behavior change.

Providers also identified a clear link between mental health support and sustained PrEP use, suggesting that mental health screening and support should be embedded in routine care. As one provider described, *“So that it can be in their minds that they really need to take PrEP… and also to monitor them accordingly and see to their mental health status continuously”* (provider, age 35). This provider also acknowledged the importance of ongoing mental health monitoring, which could reinforce adherence over time. Several providers went further, arguing that effective adherence support requires first resolving—or at least addressing—upstream issues such as gender-based and intimate partner violence, as well as the lack of family support: *“I think we will have to first illuminate the main issue which is gender-based violence by referring the patient to a social worker or police if they’re in danger. Once that has been dealt with, it’s easier for the patient to take their PrEP”* (provider, age 29). Another provider stressed the importance of including family in the intervention, recognizing that *“that’s where they get the influence of resistance from”* (provider, age 62). These perspectives collectively highlight a shared understanding that mental health is foundational to PrEP adherence. Participants and providers view psychosocial support not just as an ancillary benefit but as a core feature of any effective adherence intervention.

Intervention considerations, subtheme 2: carefully considering session format. The majority of participants and providers expressed strong preferences for mental health support delivered through group-based interventions rather than individual counseling alone. The collective sharing of experiences within a peer group was described as a powerful tool for healing, empowerment, and motivation to adhere to PrEP. Participants highlighted how hearing others’ stories and perspectives helped them gain new insights and feel less isolated. One participant reflected, *“The reason for that is that each person comes up with different views and can get advice also that I can apply in my situation. Maybe someone can share a story that is worse than yours and that can also transform my assumptions and give me strength to press through”* (pregnant, age 28). Another shared that *“sometimes when you share and listen to others it helps to forget what happened to you in the past and can guide you towards your own healing”* (pregnant, age 19), emphasizing the cathartic and normalizing effects of group dialogue.

Providers similarly advocated for group formats, recognizing that peer-to-peer support could reinforce adherence in ways professional advice alone might not. One provider explained, *“It is for them to meet with peers and share their experiences as well as to learn and support one another… when sharing the same sentiment, and have a mutual understanding… you tend to be motivated and adhere to that”* (provider, age 35). Another suggested creating specialized groups exclusively for women on PrEP with mental health issues, which could offer a more tailored and focused environment: *“Maybe have their own focus groups where they can share their own experiences”* (provider, age 28). Overall, the emphasis on session format and the clear preference for group-based programming reflect the recognized value of shared experiences and collective healing in supporting mental health and PrEP adherence among pregnant and postpartum people.

Intervention considerations, subtheme 3: recognizing and attending to the logistical challenges of integrating a mental health-focused PrEP adherence intervention into antenatal care. While participants and providers recognize the potential benefits of integrating mental health interventions within the clinic setting, several logistical challenges were highlighted. Overcrowding, limited staff capacity, and varying patient schedules complicate the practical delivery of counseling or group sessions during routine antenatal and postpartum visits. Staffing shortages and clinic congestion also pose barriers. A provider described how limited personnel during shifts lead to longer patient wait times, which can reduce patients’ willingness or ability to attend additional counseling sessions: *“By the time they must attend these sessions, they might not want to because some are hungry and tired”* (provider, age 62). Space constraints and staggered arrival times further complicate scheduling group interventions within the usual clinic flow. One provider suggested, *“Maybe if you schedule them on a different day, let them know that they’ll be attending a group session at a specific venue”* (provider, age 35), emphasizing the need for intentional planning and dedicated spaces.

To navigate these challenges, providers recommended engaging clinic leadership and staff early in the planning process. One advised, *“Firstly, discuss it with the facility manager. Introducing your goal and plans with the facility manager. They will decide if it’s okay and when you can meet up with the staff to introduce it to them too… Look at how it will benefit these women and how to work around time so they can attend these sessions”* (provider, age 62). This collaborative approach could foster buy-in and facilitate integration within existing workflows. While integrating mental health support into clinic visits presents logistical hurdles, providers’ experiences and suggestions indicate that with strategic resource leveraging—such as aligning sessions with antenatal appointments and coordinating with clinic staff—it is a feasible and potentially impactful approach to enhancing PrEP adherence.

## 3. Discussion

In this qualitative study of PPPs and healthcare workers in SA, we examined the ways in which depression and PTSD symptoms disrupt daily oral PrEP adherence during the peripartum period. Participants described how sadness, hopelessness, rumination, avoidance, and withdrawal interfered with consistent PrEP use by fueling doubts about its efficacy (often influenced by stigma and misinformation), increasing pill burden, and disrupting routines through forgetfulness and avoidance-based behaviors like hypersomnia. These symptoms also led to disengagement from healthcare providers. Given these challenges, participants expressed a strong interest in professional counseling and emphasized the need for nonjudgmental spaces offering practical support. Providers echoed the importance of integrating mental health care into routine antenatal services, including screening and counseling. Both groups supported group-based interventions for fostering peer connection and shared learning, though barriers such as staff shortages, scheduling issues, and space limitations were noted. A minority of participants preferred informal support from trusted friends or family, highlighting the need for flexible, accessible care models.

This study uniquely identifies the *specific ways* in which depression and posttraumatic stress exert their negative influence on PrEP use during pregnancy and the postpartum period, with clear intervention development implications. For PPPs with depression or PTSD symptoms, doubts about PrEP effectiveness are not only rooted in cognitive distortions common in mood and trauma disorders but also in real and perceived social consequences of PrEP use. Interventions aimed at improving adherence in this population should address PrEP-related stigma and misinformation, while also supporting individuals’ emotional regulation and decision-making under distress. Depression and trauma-related symptoms intensify the emotional and physical burden of daily PrEP use, leading to pill fatigue and exhaustion, both of which may also increase forgetfulness, not purely due to executive function but often to a manifestation of rumination, emotional overload, and unprocessed trauma. Strategies like cognitive restructuring, emotional regulation, and concrete planning strategies (e.g., pill reminders, mobile support tools, and caregiver or peer involvement) may help interrupt these ruminative cycles and reinforce PrEP use. Hypersomnia, a symptom of both depression and PTSD, may be a particularly overlooked barrier to PrEP adherence; what may appear to be “laziness” or purposeful non-engagement may be an avoidance-based coping strategy. Addressing avoidance behaviors tied to fatigue and emotional withdrawal with behavioral activation strategies and sleep hygiene education may help to disrupt maladaptive sleep patterns and rebuild the routines needed for consistent PrEP use. When symptoms lead to withdrawal from interactions with healthcare providers, PPPs may lose some of their strongest allies in their journeys to prevent HIV acquisition and transmission to their babies, indicating that social support skill-building is likely essential to any mental health-focused PrEP program.

Although studies have established robust links between mental health symptoms and decreased adherence across populations at elevated risk for HIV, few interventions have been developed to address these barriers in antenatal care. Velloza and colleagues outlined a promising approach to adapting and integrating a mental health intervention into PrEP services for adolescent girls and young women in SA, [33] yet no programs to date have focused specifically on pregnant and postpartum individuals [9]. Most participants viewed professional support as both necessary and acceptable, suggesting a strong foundation for the development of integrated interventions that pair adherence support with strategies to reduce depression and PTSD symptoms.

Participants’ preferences for group-based formats suggest that interventions grounded in shared experience may be particularly effective. Such formats can help normalize distress and foster motivation through mutual accountability and inspiration. They may be especially valuable for PPPs who experience gender-based violence or relationship inequality and need additional PrEP and emotional support, in that they may create opportunities to share strategies for navigating partner opposition, strengthening women’s sense of agency in health decision-making and building collective resilience in the face of unequal gender power dynamics.

Providers’ enthusiasm for such models—and their calls for interventions that also address structural and relational drivers of distress (i.e., gender-based violence and unsupportive family dynamics)—underscore the need for multilevel approaches that reflect the full spectrum of influences on mental health and adherence. Although providers acknowledged logistical barriers to implementation (i.e., staff shortages, time constraints, and space limitations), they offered actionable solutions, such as aligning sessions with antenatal appointments, scheduling groups on separate days, and partnering early with clinic leadership. Other provider-level data complement these findings and offer insights on strategies to support PrEP more broadly, including targeted provider training, proactive provider–patient communication, educational resources, and workflow integration strategies [34,35].

This study has several limitations. While participant interviews provided rich insights into the lived experience of PrEP use during pregnancy and postpartum, the number of provider interviews was relatively small, though they were all professional nurses (i.e., the cadre that provides most of the care in this setting). As such, the perspectives reflected may not fully capture the range of provider attitudes and experiences across different clinic settings. Limited access to clinic or health system administrators also constrained our ability to fully assess the organizational and structural barriers to implementing an integrated mental health and PrEP adherence intervention. Additionally, the sample may reflect selection bias, as individuals who agree to complete an interview could differ systematically from those who do not, particularly regarding openness about mental health experiences. Consequently, social desirability bias may have influenced participants’ responses, given the stigma around HIV, PrEP, and mental health in this context. Participants may have downplayed negative experiences with PrEP, minimized disclosure of mental health challenges, or emphasized socially acceptable coping strategies, which could limit the authenticity of reported barriers and preferences. Finally, although this study was designed to explore intervention preferences, participants were responding to hypothetical formats rather than describing their actual experiences with specific programs. This approach limited the extent to which participants could identify practical challenges such as scheduling, transportation, or competing demands. To help address this limitation, we explored potential logistical barriers in greater depth with providers, who were able to reflect on their direct experience of service delivery and clinic operations.

Despite these limitations, this study offers novel and clinically relevant insight into how depression and trauma-related symptoms disrupt PrEP use during the perinatal period, underscoring the need for contextually grounded, mental health-focused adherence interventions. PPPs with mental health challenges face intersecting emotional, cognitive, and structural barriers to daily PrEP use, yet they also articulated a clear vision for the types of support they need. Providers echoed many of these priorities, emphasizing the importance of mental health screening, peer support, and coordinated care. Addressing mental health is thus central, not ancillary, to HIV prevention in this population. These findings point to packaging evidence-based strategies that target the specific psychological disruptions identified here and implementing integrated, group-based adherence interventions tailored to perinatal life in collaboration with clinic leadership. Long-acting PrEP modalities, if found safe and acceptable during pregnancy and postpartum, could help alleviate some of the adherence burdens linked to depression and trauma. However, they may also introduce new challenges (e.g., the need for regular, time-sensitive clinic visits) which can be complicated by avoidance symptoms, or logistical barriers exacerbated by common mental health disorders. Future research should explore the appropriateness of long-acting PrEP for PPPs, facing significant mental health barriers to daily pill-taking, while ensuring that implementation strategies include adequate support to prevent new adherence challenges.

## 4. Conclusions

This study revealed that depression and PTSD symptoms contributed to missed PrEP doses or late pickups by increasing doubt about PrEP efficacy, amplifying pill burden, intensifying avoidance and withdrawal (e.g., hypersomnia and disengagement from providers), and disrupting memory through rumination and emotional overload; most PPPs preferred support from professional counselors, while a minority preferred informal support; and intervention design considerations included aligning patient and provider goals, selecting between individual and group formats, and addressing integration barriers such as staffing and space constraints. Providers additionally affirmed the need for mental health support in antenatal care. Intervention strategies that focus on increasing PrEP knowledge and motivation while targeting emotional withdrawal, fatigue, and cognitive overload may improve adherence and reduce HIV risk in this population.

## Figures and Tables

**Figure 1 ijerph-22-01350-f001:**
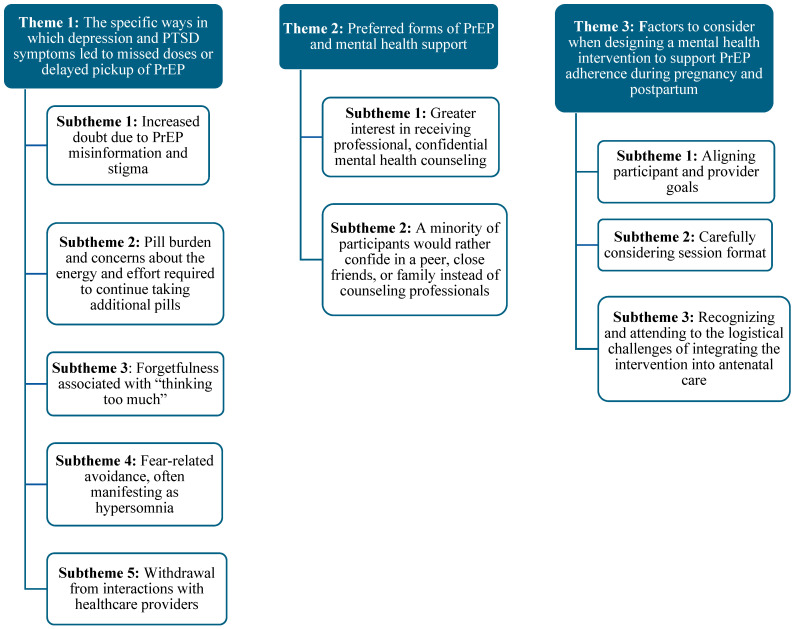
Qualitative interview themes and subthemes.

**Table 1 ijerph-22-01350-t001:** Participant demographics.

	M ± SD	N (%)
Age	28.9 ± 7.4	
Gestational age in weeks (*n* = 10 pregnant participants)	25.9 ± 8.0	
Weeks postpartum (*n* = 18 postpartum participants)	9.4 ± 11.2	
Number of times given birth to a live baby	1.8 ± 1.1	
**Race:**		
Black South African		24 (85.7)
Black Non-South African		2 (7.1)
**Highest Level of Schooling:**		
Grade 7/Std 5		1 (3.6)
Grade 9/Std 7		2 (7.1)
Grade 10/Std 8		3 (10.7)
Grade 11/Std 9		10 (35.7)
Grade 12/Std 10		7 (25.0)
Tertiary, University		3 (10.7)
**Employment Status:**		
Full-time		3 (10.7)
Self-employed		1 (3.6)
Other		1 (3.6)
Not employed		21 (75.0)
**Household Monthly Income:**		
0 ZAR		6 (21.4)
Less than 3000 ZAR (~USD 170)		14 (50.0)
3001–6000 ZAR (~USD 170–USD 340)		4 (14.3)
6001–9000 ZAR (~USD 340–USD 510)		1 (3.6)
**PrEP Status:**		
Recent PrEP initiation (<1 month) per medical chart review		16 (57.1)
PrEP adherence per medical chart review		8 (28.6)
Self-reported PrEP adherence challenges		23 (82.1)
EPDS total score	16.1 ± 4.5	
PCL-5 total score	42.8 ± 15.2	
**Elevated symptoms:**		
Depression and PTSD		16 (57.1)
Depression alone		11 (39.3)
PTSD alone		1 (3.6)

**Table 2 ijerph-22-01350-t002:** Provider demographics.

Age, M (SD)	41.7 (15.7)
Race, N (%)	
Black South African	6 (100%)
Gender, N (%)	
Cisgender woman	6 (100%)
Education, N (%)	
University	6 (100%)
Healthcare Role, N (%)	
Professional nurse	6 (100%)
Years in healthcare, M (SD)	14.8 (13.2)
Years in current role, M (SD)	8.3 (8.5)
Years at current clinic, M (SD)	11.2 (12.8)
Years in antenatal care, M (SD)	8.3 (11.9)

## Data Availability

The original contributions presented in this study are included in the article. Further inquiries can be directed to the corresponding author.

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
