# Peer review of "“I’ll Continue If I Have a Positive Mind”: Identifying the Ways in Which Depression and PTSD Impact PrEP Adherence Among PrEP-Experienced Pregnant and Postpartum Women in Cape Town, South Africa"

_ijerph, 2025, doi:10.3390/ijerph22091350_

Round 1

Reviewer 1 Report

Comments and Suggestions for Authors

Congratulations to the Authors for conducting such an important study and for the clear and engaging presentation of the findings. The manuscript is well-written and offers valuable insights that can inform future mental health and PrEP adherence interventions.

My suggestions aim to further enhance clarity, deepen the discussion, and strengthen the research's impact. Please see below:

Title and Abstract: The title is captivating and accurately reflects the study's focus. The abstract is informative and concisely presents the main findings. Suggestion: The abstract mentions "three key findings," but Figure 1 details five subthemes for "Theme 1." While the three findings in the abstract refer to the general themes (symptom impact, support preferences, design considerations), a slight rephrasing could make the correspondence more explicit or clarify that the subthemes are deeper explorations of the findings. For example, specify that the first finding "revealed three key findings: (1) depression and PTSD symptoms contributed to missed PrEP doses or late pickups by increasing doubt about PrEP efficacy, amplifying pill burden, intensifying avoidance and withdrawal (e.g., hypersomnia, disengagement from providers), and disrupting memory through rumination and emotional overload" comprises the five mechanisms detailed in Figure 1.

Introduction: The introduction excellently contextualizes the high HIV acquisition rate among pregnant and postpartum people (PPP) in Sub-Saharan Africa and the importance of PrEP, as well as adherence challenges. The gap in the literature on how depression and PTSD specifically impact PrEP adherence during pregnancy and how to address them in low-resource settings is well justified.

Strong Point: The introduction highlights the complexity of the problem, citing biological, structural, and, crucially, psychological factors, such as intimate partner violence (IPV) and its impacts on PTSD, as per lines 64-77. The integration of these multiple layers is a strength of the study.

Strong Point: The presentation of PrEP efficacy and known adherence challenges (lines 51-63) sets the stage for the exploration of mental health factors.

Methods: The methods section is detailed and transparent.

Study Procedures:

Recruitment and Eligibility: The description of the recruitment process for PPP and providers is clear. Prenatal care clinics as recruitment sites are logical.

Eligibility Criteria: The criteria for PPP are well-defined, including cutoff scores for EPDS (≥ 11) and PCL-5 (≥ 31) for elevated depression and PTSD symptoms, respectively. Footnote 28 regarding the legal age for reproductive health decisions in South Africa is important for understanding the consent/assent process.

Sample: The number of participants (28 PPP and 6 providers) is adequate for an in-depth qualitative study. Suggestion: In Section 3 (Results), the most common reasons for ineligibility are presented as follows: "Most common reasons for ineligibility included not meeting criteria for elevated depression or PTSD symptoms (95% of total ineligible, n = 35), not self-reporting adherence challenges (84% of total ineligible, n = 31), and no delayed PrEP pickup (78% of total ineligible, n = 29)." The sum of these percentages exceeds 100%, indicating that the categories are not mutually exclusive, and a single individual may have fallen into multiple reasons for ineligibility. It would be useful to clarify in the Methods or Results Section that these are prevalent reasons and not exclusive categories, to avoid confusion about the total count of ineligible participants.

Analysis:

The description of the thematic analysis process (for PPP) and rapid qualitative analysis (for providers) is thorough. Team coding, discrepancy resolution, and consultation with the South Africa-based team for cultural nuances are practices that reinforce the validity of the findings.

Strong Point: The reference to "Tracy's eight criteria for high-quality qualitative research" (lines 184-200) is an excellent detail, demonstrating rigor and reflexivity. Detailing how the study addressed each criterion (worthy topic, rich rigor, sincerity, credibility, resonance, significant contribution, ethical standards, and meaningful coherence) is exemplary and serves as a model for other qualitative studies.

Results: The presentation of the results is logically organized, with participant and provider demographic tables, followed by the articulation of themes and subthemes. Figure 1 is an excellent visual summary of the qualitative findings.

Demographics: Tables 1 and 2 provide a comprehensive overview of the sample characteristics.

Themes and Subthemes:

Theme 1: The impact of depression and PTSD symptoms on PrEP non-adherence. This theme is the core of the study and is explored in great depth and richness of detail, supported by direct and compelling quotes from participants and providers.

Subtheme 1: Increased doubt due to PrEP misinformation and stigma. The interrelationship between emotional state ("When I was feeling sad, I would start doubting...") and stigma ("Some people assume that it’s ARVs and don’t know what the pills are for...") is well captured and crucial.

Subtheme 2: Strong sense of pill burden and concerns about the energy and effort required. Quotes like "I will see once I have given birth but it’s not nice drinking pills every day." and "Sometimes I would not take it [PrEP] because I was not in the mood for anything" vividly illustrate "pill fatigue" and disengagement. The participant's quote about being a rape victim and the burden of multiple medications ("I am a victim of 33 rapes. So, I wasn’t only taking PrEP. I took a lot of medication...") is particularly powerful and adds a critical dimension to understanding adherence in trauma contexts.

Subtheme 3: Forgetfulness associated with "overthinking." The connection between rumination ("I used to think a lot about other things rather than taking PrEP") and forgetfulness is an important insight, showing that it is not mere inattention, but cognitive overload.

Subtheme 4: Avoidance related to fear manifesting as hypersomnia. Hypersomnia as a coping mechanism is a distinctive finding and deserves emphasis. "I prefer to sleep so that I do not come into contact with other people…so that they cannot be affected by my bad mood..." demonstrates the complexity of this behavior.

Subtheme 5: Withdrawal from interactions with healthcare providers. The "overwhelming reluctance to speak with clinic staff" and negative experiences with nurses are critical for understanding barriers in the healthcare system.

Theme 2: Preferred forms of PrEP and mental health support.

Subtheme 1: Increased interest in receiving professional and confidential counseling. The preference for the neutrality and confidentiality of professional counseling ("I will be free and will be easy for me to speak without feeling being judged...") is a strong driver for intervention design.

Subtheme 2: Minority of participants preferring to confide in peers, intimate friends, or family. The value placed on mutual connection and understanding ("preferably a peer because there can be a mutual connection...") highlights the importance of informal social support. The quote about logistical barriers to formal counseling ("No, because I do not even know where to go for it and how accessible is it...") is fundamental for planning accessibility.

Theme 3: Factors to consider in designing a mental health intervention.

Subtheme 1: Shared recognition of mental health as a basis for PrEP adherence. The idea that "effective adherence support requires first resolving—or at least addressing—upstream issues such as gender-based and intimate partner violence" is an important call for an integrated approach.

Subtheme 2: Careful consideration of session format. The clear preference for group interventions ("each person comes up with different views and and can get advice also that I can apply in my situation. Maybe someone can share a story that is worse than yours...") is a valuable insight for intervention design.

Subtheme 3: Recognition and attention to the logistical challenges of integrating the intervention. Providers articulate well the practical barriers, such as overcrowding and staff shortages ("By the time they must attend these sessions, they might not want to because some are hungry and tired") and offer pragmatic solutions such as "schedule them on a different day."

Strong Point: The alternation between PPP and provider perspectives throughout the results enriches the analysis and offers a more complete view of the proposed challenges and solutions. Suggestion: Given the richness of the data, the authors could consider how to illustrate the interconnection of the subthemes even more explicitly in the text (beyond Figure 1), perhaps with an introductory paragraph for each theme that highlights how the different aspects complement or exacerbate each other.

Discussion: The discussion synthesizes the findings well and places them in context with existing literature.

Implications for Intervention Development: This is one of the strongest sections, as it translates qualitative findings into actionable recommendations. The emphasis on addressing stigma, misinformation, emotional regulation, and concrete planning strategies is crucial. The discussion of hypersomnia as a neglected barrier ("what may appear to be “laziness” or purposeful non-engagement may be an avoidance-based coping strategy") is a particularly insightful finding and deserves to be emphasized. Strong Point: The recognition that "Address mental health is thus central, not ancillary, to HIV prevention in this population" is a powerful and well-founded message.

Strong Point: The discussion of long-acting PrEP modalities is pertinent and shows the authors' prospective vision, including consideration of how mental health symptoms may impact these new modalities.

Limitations: The limitations are adequately addressed, including the provider sample size, potential selection bias, and the hypothetical nature of intervention preferences.

Suggestion: It could be interesting to discuss the potential influence of social desirability on participants' responses, given the sensitive nature of the topic (HIV, PrEP, mental health, violence). Although researchers created a safe environment, it is a factor inherent to interviews on such personal topics. Suggestion: The discussion could explore a bit more the gender and power implications in PrEP adherence, especially considering the mentions of IPV and gender inequity in the introduction and the importance of family support in Section 3. Although the topic has been indirectly addressed, a more explicit discussion of how interventions can empower women in this context would be valuable.

Conclusion: The conclusion is concise and reaffirms the study's central message.

References: The references are comprehensive and appropriate, supporting the claims made in the text.

Quality of Writing and Clarity: The manuscript is exceptionally well-written. The language is clear and accessible, and the logical flow of the sections facilitates understanding of the complex findings. The translation into English is well done, and the quotes are impactful.

In summary, this is a highly meritorious study that offers profound and actionable insights for the field of HIV prevention and mental health. The suggestions presented are only aimed at refining an already excellent work.

Author Response

Comment 1: The abstract mentions "three key findings," but Figure 1 details five subthemes for "Theme 1." While the three findings in the abstract refer to the general themes (symptom impact, support preferences, design considerations), a slight rephrasing could make the correspondence more explicit or clarify that the subthemes are deeper explorations of the findings. For example, specify that the first finding "revealed three key findings: (1) depression and PTSD symptoms contributed to missed PrEP doses or late pickups by increasing doubt about PrEP efficacy, amplifying pill burden, intensifying avoidance and withdrawal (e.g., hypersomnia, disengagement from providers), and disrupting memory through rumination and emotional overload" comprises the five mechanisms detailed in Figure 1.

Response to feedback: Thank you for this feedback. We have integrated some language in this sentence of the abstract to clarify that the subthemes are more specific mechanisms or details of the broader key finding (see page 1: “Thematic analysis revealed three key findings with subthemes that deepen exploration of each theme:…”

Comment 2: In Section 3 (Results), the most common reasons for ineligibility are presented as follows: "Most common reasons for ineligibility included not meeting criteria for elevated depression or PTSD symptoms (95% of total ineligible, n = 35), not self-reporting adherence challenges (84% of total ineligible, n = 31), and no delayed PrEP pickup (78% of total ineligible, n = 29)." The sum of these percentages exceeds 100%, indicating that the categories are not mutually exclusive, and a single individual may have fallen into multiple reasons for ineligibility. It would be useful to clarify in the Methods or Results Section that these are prevalent reasons and not exclusive categories, to avoid confusion about the total count of ineligible participants.

Response to feedback: Thank you for pointing out that this breakdown of ineligibility was confusing. We have added some language to clarify that individuals may have met multiple exclusion criteria (see page 8): Most common reasons for ineligibility included not meeting criteria for elevated depression or PTSD symptoms (95% of total ineligible, n = 35), not self-reporting adherence challenges (84% of total ineligible, n = 31), and no delayed PrEP pickup (78% of total ineligible, n = 29); these reasons were not mutually exclusive, with many individuals meeting multiple criteria for exclusion.”

Comment 3: It could be interesting to discuss the potential influence of social desirability on participants' responses, given the sensitive nature of the topic (HIV, PrEP, mental health, violence). Although researchers created a safe environment, it is a factor inherent to interviews on such personal topics.

Response to feedback: Thank you for this suggestion. We have added a brief description in the limitations section (pages 26-27) to discuss the potential of social desirability bias given the stigma around HIV, PrEP, and mental health in this context, “Consequently, social desirability bias may have influenced participants’ responses given the stigma around HIV, PrEP, and mental health in this context. Participants may have downplayed negative experiences with PrEP, minimized disclosure of mental health challenges, or emphasized socially acceptable coping strategies, which could limit the authenticity of reported barriers and preferences.”

Comment 4: The discussion could explore a bit more the gender and power implications in PrEP adherence, especially considering the mentions of IPV and gender inequity in the introduction and the importance of family support in Section 3. Although the topic has been indirectly addressed, a more explicit discussion of how interventions can empower women in this context would be valuable.

Response to feedback: Thank you for this suggestion. We have added some text on page 25 to highlight the ways in which a group-based intervention format may provide a safe and shared space for PPP who are experiencing gender-based violence or relationship inequality, given the influence of gender power dynamics on PrEP use in South Africa, “Such formats can help normalize distress and foster motivation through mutual accountability and inspiration. They may be especially valuable for PPP who experience gender-based violence or relationship inequality and need additional PrEP and emotional support, in that they may create opportunities to share strategies for navigating partner opposition, strengthening women’s sense of agency in health decision-making, and building collective resilience in the face of unequal gender power dynamics.”

Reviewer 2 Report

Comments and Suggestions for Authors

This is a well-written paper that explores the impact of depression and PTSD symptoms on adherence to pre-exposure prophylaxis (PrEP) among pregnant and postpartum women (PPP) in Cape Town, South Africa. The methodology is apt, and the findings, discussions and implications are well discussed.

Overall this is a well-structured and relevant study. I have only a few minor suggestions for the author. They are given below:

A. Please be mindful about spacing. There is a big gap between page 3 and page 4.

B. The initial recruitment happened when the participants were approached by study research assistants while waiting in line at the antenatal clinic. Can this process be further explained? Approaching participants randomly in a clinical setting can be complicated and compromise patient confidentiality and identity. Was this part of the IRB application and received approval from IRB? If you could explain this further, and the steps you took to protect patient/participant confidentiality that would be great.

C. Please check all acronyms that you have used and ensure all have been spelled out when used initially.

D. Was the table (figure 1) on page 8 created with the Dedooce software?

E. The last limitation " although the study was designed to explore intervention 632
preferences, participants were responding to hypothetical formats rather than describing their experiences with specific programs" on page 15 needs to be explained better in the discussion section.

Author Response

Comment 1: Please be mindful about spacing. There is a big gap between page 3 and page 4.

Response to feedback: Thank you for your comment. However, we do not see the spacing problem on our end. This may have been a problem when the manuscript was downloaded. We will ensure that this is resolved with the editor.

Comment 2: The initial recruitment happened when the participants were approached by study research assistants while waiting in line at the antenatal clinic. Can this process be further explained? Approaching participants randomly in a clinical setting can be complicated and compromise patient confidentiality and identity. Was this part of the IRB application and received approval from IRB? If you could explain this further, and the steps you took to protect patient/participant confidentiality that would be great.

Response to feedback: Thank you for bringing this point up. The research assistants did not ask patients for any identifying information while they were in queue. Patients were just informed of the study opportunity, and, if they expressed interest, they were brought to a private space for the consent process and eligibility screener to ensure patient confidentiality. Providers were approached outside of working hours to protect their clinic responsibilities and their privacy. These recruitment procedures were outlined in the study protocol and were approved by the local IRB, which in South Africa is called the Human Research Ethics Committee (HREC). We have added text to pages 4-5 to clarify recruitment procedures, “Research assistants briefly described the study to PPP, and those who expressed interest in completing an interview were invited to speak to a research assistant in a private space at the clinic to be consented and screened for eligibility. Research assistants did not ask potential participants any personal questions while in queue to ensure confidentiality.”

Comment 3: Please check all acronyms that you have used and ensure all have been spelled out when used initially.

Response to feedback: Thank you for pointing us to this. All acronyms have been checked to ensure they are fully spelled out in their initial iteration.

Comment 4: Was the table (figure 1) on page 8 created with the Dedooce software?

Response to feedback: Thank you for your question. The figure was not made with Dedoose, it was made by us with Microsoft Word.

Comment 5: The last limitation "although the study was designed to explore intervention
preferences, participants were responding to hypothetical formats rather than describing their experiences with specific programs" on page 15 needs to be explained better in the discussion section.

Response to feedback: Thank you for this comment. We have revised the limitation to clarify that, because participants were asked to reflect on to hypothetical intervention formats, their responses may not fully capture the practical or logistical barriers that they might face if participating in an actual program. As a result, their preferences emphasized perceived benefits rather than actual, experienced benefits. We supplemented these perspectives with provider interviews, which offered concrete insights into the logistical and organizational issues that might arise during intervention implementation and delivery (see page 27), “Finally, although the study was designed to explore intervention preferences, participants were responding to hypothetical formats rather than describing their actual experiences with specific programs. This approach limited the extent to which participants could identify practical challenges such as scheduling, transportation, or competing demands. To help address this limitation, we explored potential logistical barriers in greater depth with providers, who were able to reflect on their direct experience of service delivery and clinic operations.”